# Effect of Heat Treatment on the Cyclic Deforming Behavior of As-Extruded ZA81M Magnesium Alloy

Tianjiao Luo [1,*], Jianguang Feng [1], Chenye Liu [1], Cong Wang [2], Yingju Li [1], Xiaohui Feng [1], Ce Zheng [1], Qiuyan Huang [1], Weirong Li [3] and Yuansheng Yang [1,4,*]

[1] Institute of Metal Research, Chinese Academy of Sciences, Shenyang 110016, China; jgfeng19b@imr.ac.cn (J.F.); cyliu20s@imr.ac.cn (C.L.); yjli@imr.ac.cn (Y.L.); xhfeng@imr.ac.cn (X.F.); czheng14b@imr.ac.cn (C.Z.); qyhuang16b@imr.ac.cn (Q.H.)

[2] School of Material Science and Engineering, North University of China, Taiyuan 038507, China; 20210033@nuc.edu.cn

[3] Dongguan Eontec Co., Ltd., Dongguan 523662, China; liwr@e-ande.com

[4] Shandong Provincial Key Laboratory of High Strength Lightweight Metallic Materials, Advanced Materials Institute, Qilu University of Technology (Shandong Academy of Sciences), Jinan 250014, China

[*] Correspondence: tjluo@imr.ac.cn (T.L.); ysyang@imr.ac.cn (Y.Y.); Tel.: +86-24-23971109 (T.L.); +86-24-23971728 (Y.Y.)

**Abstract:** In the present work, the effect of heat treatment on the cyclic deformation behavior of as-extruded ZA81M magnesium alloy was investigated. Two heat treatment conditions were applied to the as-extruded ZA81M alloy: a solution treatment (T4, 653 K for 40 h and quenched with 298 K water) and a solution treatment plus artificial aging (T6, 348 K for 32 h (pre-aging at low temperature) and 453 K for 8 h (the second aging) and quenched with 353 K water). The results showed that the fine second phase precipitated after the aging treatment, the tensile yield strength of the T6-treated specimens increased, and the stress amplitude of T6-treated specimens was always higher than that of T4-treated specimens. The T6-treated specimens had a higher total strain energy density and a shorter fatigue life at a strain amplitude of 0.4%, and a lower total strain energy density and a longer fatigue life at a strain amplitude of 0.8%, compared to the T4-treated specimens. All fatigue cracks of the T4 and T6 ZA81M alloy were initiated at the second phase or along the grain boundary and propagated perpendicular to the loading direction.

**Keywords:** ZA81M magnesium alloy; heat treatment; cyclic deformation; fatigue life; hysteresis loop



## 1. Introduction

Magnesium alloys, as the lightest metal structural materials, are important lightweight materials used in the transportation, electronic communications, aerospace, and defense industries [1–3]. They have been successfully applied for non-bearing structures, and where there is a requirement for light weight, now the application of magnesium alloys is gradually extending from non-load-bearing parts to load-bearing structures. Because structures not only have to bear static loads but also need to bear dynamic loads, magnesium alloys must not only have a high strength and toughness, but also a high fatigue resistance. In order to improve the service reliability and fatigue life of structural components and devices, the cyclic deformation behavior of magnesium alloys needs to be investigated. Previous studies have mainly focused on the effects of twinning and texture on the fatigue properties of magnesium alloys [4–6]. Lee et al. [4] observed the cyclic deformation behavior in situ by neutron diffraction and reported that, for tension followed by reverse compression, little detwinning occurs after the initial tension stage, but almost all of the twinned volumes are detwinned during loading in reverse compression. Wu et al. [5] discovered that, for an extruded AZ31 plate in a low cycle fatigue test under a tension-tension load, twinning-dominated samples showed more pronounced cyclic hardening and longer fatigue life than those of slip-dominated samples. The elongated lifetime of twinning-dominated samples may be due to roughness-induced crack closure,

but Jordon et al. [6] reported that the inclusion size was more important in determining the fatigue life than the anisotropic effects from the texture, yield, and work hardening. The effect of heat treatment on the fatigue of magnesium alloys has also been covered in previous studies [7–11], Zapletal et al. [7] conducted the heat treatment of AZ61 alloy and studied the cyclic deformation and fatigue life of the optimal heat treatment condition; they found that heat treatment had no significant effect on the low- and high- cycle fatigue behavior. However, Mehedi and Mirza et al. [8–11] reported that heat treatment could significantly affect the fatigue properties of a magnesium alloy containing rare earth elements, mainly because high-density nano precipitates could be precipitated during heat treatment.

ZA81M is a recently developed high-strength and ductility free-RE magnesium alloy developed by our team [12–15]; it has good application prospects. Wang et al. [12] investigated the microstructure and mechanical property of as-cast ZA81M alloy and found that optimized heat treatment led to improved performance—i.e., the yield strength (YS), ultimate tensile strength (UTS), and elongation of T6 state alloy reached 228 MPa, 328 MPa, and 16.0%, which are much higher than the values achieved by T4 state alloy. In addition, Zhu et al. [13,14] found that the ZA81M alloy not only had good casting properties but also good deformation performance. The tensile strength of as-extruded ZA81M alloy was enhanced remarkably by solution + double-aging heat treatment, with the yield strength, tensile strength, and elongation becoming 298 MPa, 348 MPa, and 18%, respectively. And Wang et al. [15] investigated the anisotropic cyclic deformation behavior of extruded ZA81M magnesium alloy with a tilted basal texture and found that there were obvious asymmetries in the cyclic deformation along extrusion direction and transverse direction. Our investigations into this alloy indicated that because numerous high-density nano precipitates formed in the ZA81M alloy during heat treatment, the tensile properties of as-cast and as-extruded ZA81M alloy were markedly improved. However, there is a lack of research on the effect of heat treatment on the cyclic deformation behavior of ZA81M alloy, especially on the mechanism of the effect of nanoprecipitation on the fatigue failure of ZA81M alloy.

In this study, the cyclic deformation behavior of T4 (solution treated at 653 K for 40 h and quenched with 298 K water) and T6 (pre-aging 348 K for 32 h and the second aging 453 K for 8 h and quenched with 353 K water) samples machined from an as-extruded ZA81M magnesium alloy bar was investigated and an analysis of T4 and T6 samples was carried out in order to clarify the role of the heat treatment in the initiation and propagation of fatigue cracks, and reveal the mechanism of nano precipitate in fatigue failure.

## 2. Materials and Methods

### 2.1. Materials and Conditions of Heat Treatment

Table 1 shows the chemical composition of the ZA81M magnesium alloy used in this study. First, the alloy was melted from pure Mg, pure Zn, pure Al, pure Cu, and Mg-10 Mn (wt.%) master alloy in an electric resistance furnace then the melt was poured into a direct chill (DC) casting machine (Xishan Transformer Electric Furnace Factory, Wuxi, China) in order to cast an ingot with a diameter of 150 mm, in which the casting speed, melt temperature, and water flow were kept constant at 150 mm/min, 1003 K, and 30 L/min, respectively. After DC casting, the ingots were homogenized at 633 K for 60 h and extruded into bars with a diameter of 20 mm at 290 °C with an extrusion ratio of 39 and a ram speed of 0.5 mm/s.

**Table 1.** Chemical compositions of the ZA81M magnesium alloy (wt. %).

| Zn | Al | Cu | Mn | Mg |
|-----|-----|-----|-----|-----|
| 7.7 | 0.9 | 0.4 | 0.5 | Bal. |

The samples used for microstructure examination were machined from the bar using an electrical discharged machine and ground using SiC paper from 800 grit up to 2000 grit, followed by polishing them using 2.5 μm diamond paste. An etchant containing

4.2 g picric acid, 10 mL acetic acid, 10 mL $H_2O$, and 70 mL ethanol was used to reveal the microstructure.

The extrusion bars were solution-treated (T4) at 653 K for 40 h and quenched with 298 K water. After solution treatment, the bars were peak-aged (T6) at 348 K for 32 h (pre-aging at low temperature) and 453 K for 8 h (the second aging) and quenched with 353 K water.

Round tensile test specimens following the ASTM E8 standard with a diameter of 3 mm and dog bone-shaped fatigue specimens with a cross section area of $4 \times 4$ mm$^2$ within the gauge section were machined from the extrusion bar along ED using an electrical discharged machine (Taizhou Ruite Machinery Equipment Co., Ltd, Taizhou, China). The gauge section of the fatigue specimens was ground using SiC paper of 5000 grit and polished using 2.5 μm diamond paste. The specimen shapes are shown in Figure 1.

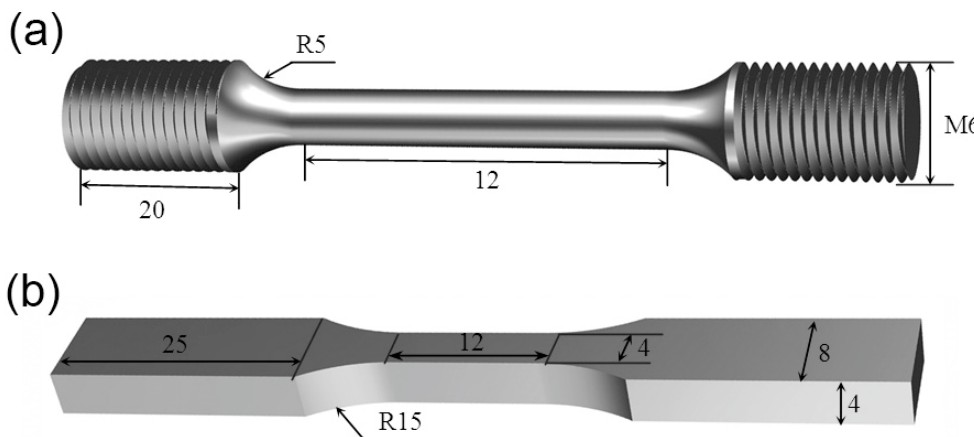

**Figure 1.** Shapes of specimens (mm): (**a**) tensile test specimen; (**b**) fatigue specimen.

### 2.2. Experimental Procedures

Tensile tests were carried out on an AG-100 kNG universal tensile testing machine (Shimadzu, Chengdu, China) at a constant strain rate of $1 \times 10^{-3}$ s$^{-1}$ at room temperature in air. Fatigue tests were conducted with a strain rate of $1 \times 10^{-2}$ s$^{-1}$, strain ratios of S = −1 and S = −3, and strain amplitudes of 0.4% and 0.8% using fully reversed push-pull mode on an Instron Electropuls E10000 servo hydraulic testing machine (Instron, Norwood, MA, USA) at room temperature in laboratory air. An extensometer with a gauge length of 10 mm was used to measure the strain.

The microstructures were analyzed using a ZEISS optical microscope (Zeiss, Oberkochen, Germany), a JSM-6460 scanning electron microscope (SEM, JEOL, Akishima-shi, Japan), and a JEM-2010 transmission electron microscope (TEM, JEOL, Akishima-shi, Japan). The fatigue fracture surfaces were observed with the JSM-6460 SEM.

## 3. Results and Discussion

### 3.1. Microstructure and Mechanical Properties after T4 and T6 Heat Treatment

Figure 2 shows the optical microstructure of the alloy after T4 and T6 heat treatment. Equiaxed grains of about 15 μm were observed under both conditions, and there were some large second phases that were similar in both the T4- and T6-treated alloys. In Figure 3, the XRD patterns indicate that the alloy in T4 and T6 states are composed of $\alpha$-Mg, MgZn$_2$, and MgZnCu phases, as demonstrated by the EDS results. These results show that A contained Mg and Zn and B contained Mg, Zn, and Cu. The phases were similar to those in the as-cast ZA81M alloy [12].

Figure 4 shows the bright field TEM images of T4 and T6 specimens, which suggest that more small second phases were precipitated during the T6 heat treatment than during the heat treatment of the T4 specimen, and the precipitated nano-phase was rod-shaped

with its long axis parallel to the $[0001]_\alpha$ of the $\alpha$-Mg matrix. The TEM analysis pattern showed that the rod-shaped phases were $MgZn_2$.

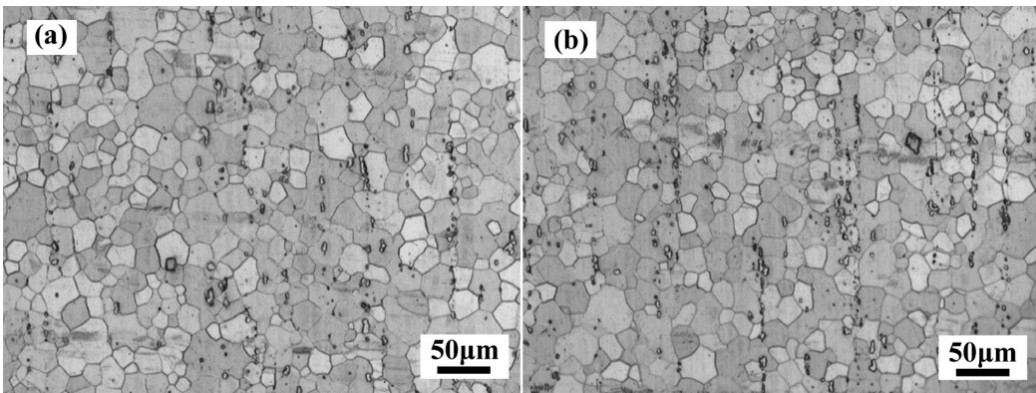

**Figure 2.** Optical micrograph of ZA81M magnesium alloy under (**a**) T4 and (**b**) T6 heat treatment.

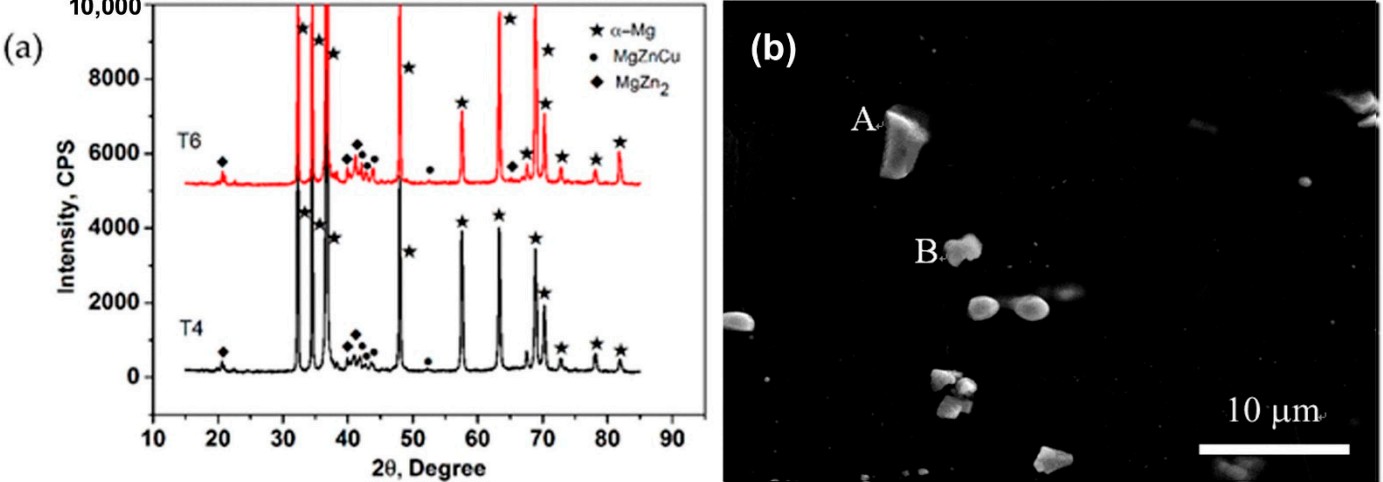

**Figure 3.** (**a**) XRD patterns of extruded ZA81M magnesium alloy under T4 and T6 heat treatment; (**b**) SEM image showing the testing points of the EDS of the extruded ZA81M magnesium alloy.

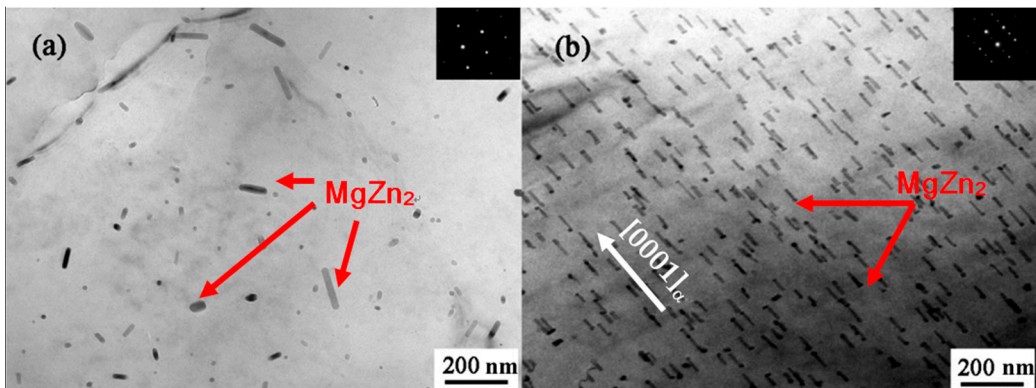

**Figure 4.** Bright field TEM images of ZA81M under (**a**) T4 and (**b**) T6 heat treatment.

Figure 5 shows the engineering stress–strain curves of the alloy under T4 and T6 heat treatments. The tensile yield strength, ultimate tensile strength, and elongation under the T4 heat treatment were 208 MPa, 330 MPa, and 26.5%, respectively. Tensile yield strength was increased to 296 MPa and the elongation was decreased to 18.5% after subsequent

aging treatment, which gave a similar result to the ZK60 magnesium alloy subjected to T4 and T6 heat treatments [16]. The enhancement of the tensile yield strength was mainly the result of the precipitation of the fine second phase acting as a barrier to dislocation glide on the prismatic plane, which is the main slip system of extruded magnesium alloy exhibiting a basal texture during tension along the extrusion direction [17].

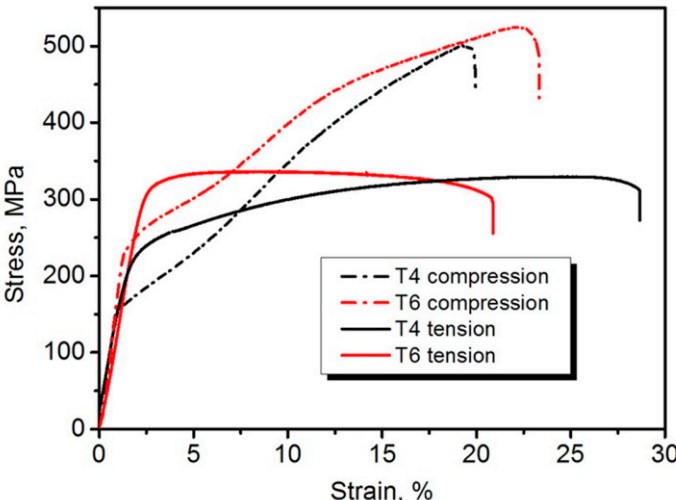

**Figure 5.** Engineering strain–stress curves of ZA81M under T4 and T6 heat treatment.

### 3.2. Cyclic Stress Response

Figure 6 shows the stress amplitude of T4 and T6 specimens under strain amplitudes of 0.4% and 0.8%. Here, the stress amplitudes of T6 specimens were higher than those of T4 specimens at the beginning of the cyclic deformation, which was the result of the higher tensile yield strength of the T6 specimens. Cyclic hardening was detected for both kinds of specimens, with T4 specimens showing higher cyclic hardening rates under the same strain amplitude. However, the stress amplitudes of T6 specimens are always higher.

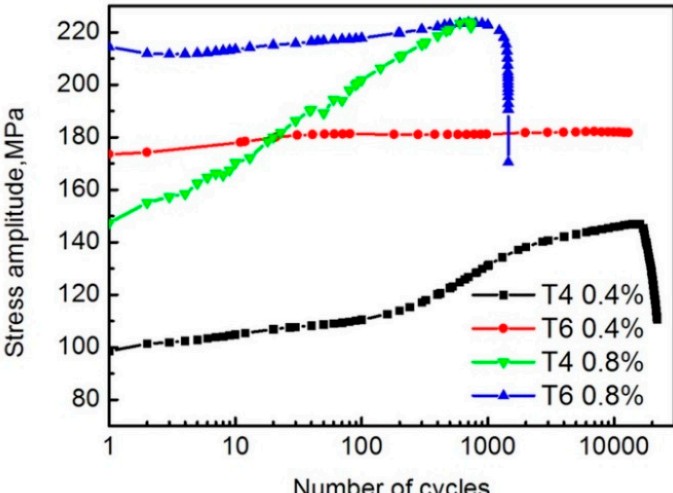

**Figure 6.** Cyclic stress response of the T4- and T6-treated ZA81M magnesium alloys.

Tensile peak stresses and compressive peak stresses are shown in Figure 7; both increased with an increase in the number of cycles. When the strain amplitude increased from 0.4% to 0.8%, the compressive peak stresses increased for the T4 specimens but remained constant for the T6 specimens. The increase in the dislocation density and their interaction contributed to the cyclic hardening [18].

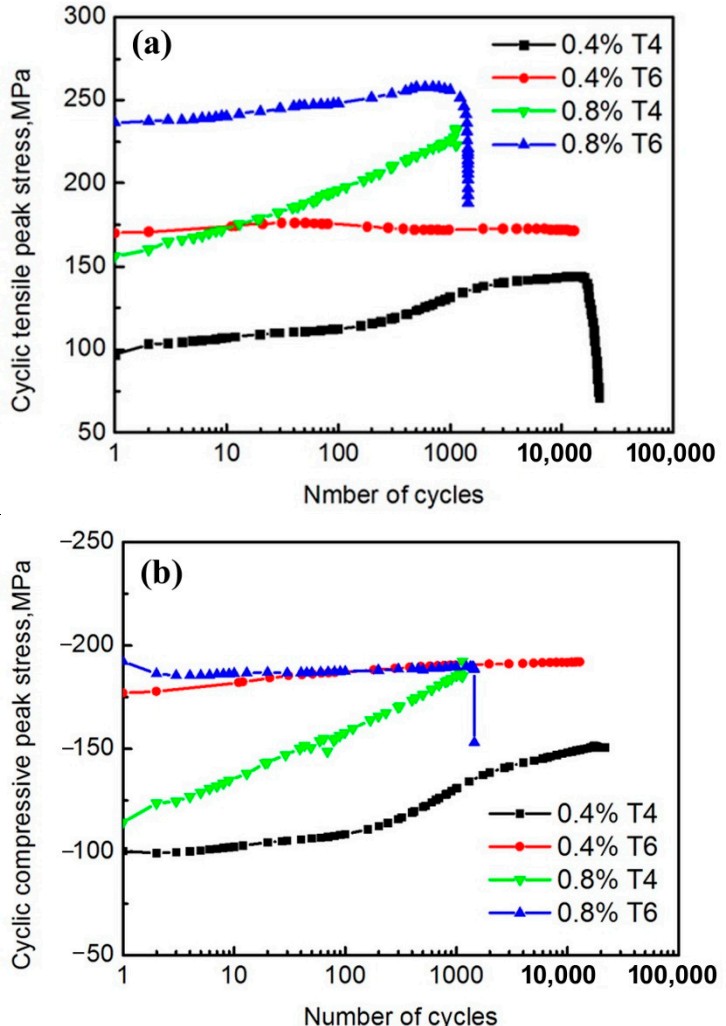

**Figure 7.** Evolvement of cyclic tensile peak stress (**a**) and cyclic compressive peak stress (**b**) with increasing numbers of cycles.

The total strain amplitude is the sum of the elastic strain amplitude and the plastic strain amplitude—i.e., $\Delta\varepsilon_t = \Delta\varepsilon_e + \Delta\varepsilon_p$. Since the yield strength of the T6 specimens was higher than that of the T4 specimens, as shown in Figure 5, under the same total strain amplitude the elastic strain amplitude of T6 specimens was bigger and the plastic strain amplitude was smaller, which did not favor strain hardening, because strain hardening is related to the dislocation multiplication during deformation and the dislocation multiplication rate under a small strain amplitude is slow. The plastic strain amplitude of the T4 specimens under a strain amplitude of 0.4% was less than that of T6 specimens under a strain amplitude of 0.8%, but the T4 specimens still exhibited a higher hardening rate; therefore, there may be other factors affecting the strain hardening rate. Yakubtsov et al. [19] found that aging treatment decreased the hardening rate of AZ80 magnesium alloy under tensile deformation and suggested that the alloying element decreased during the aging treatment, which then increased the stacking fault energy, increased the chance for dislocation, and led to cross slipping and them destroying each other. Chen et al. [20] found that T4 heat treatment increased the hardening rate, while the T5 and T6 heat treatments decreased the hardening rate of ZK60 magnesium alloy. They ascribed this to the effect of the second phase acting as a dislocation sink. Dong et al. [21] also reported a lower hardening rate of ZK60 magnesium alloy after aging treatment. Thus, the lower hardening rate of T6-treated specimens was the result of a lower content of the solution element and the precipitation of the second phase.

*3.3. Hysteresis Loops*

Figure 8 shows the first cycle of hysteresis loops for T4- and T6-treated specimens. Under a strain amplitude of 0.4%, asymmetric hysteresis loops were observed for the T4-treated specimen with a strain hardening plateau in the compressive stage of the cycle. The T6 specimens showed symmetric hysteresis loops, and it was found that the tensile peak stress was under 200 MPa, indicating that no plastic deformation had happened. Under a strain amplitude of 0.8%, a hardening plateau was observed in both specimens, indicating that twinning deformation happened during the compressive stage (work hardening), which resulted in asymmetric hysteresis loops, but the T6 specimens exhibited less asymmetric hysteresis loops. The strain hardening plateau resulted from twinning deformation [22,23], so the aging treatment increased the stress for the onset of twinning and decreased the extent of twinning deformation [24]. Stanford et al. [25] found that the fine second phase in Mg-5Zn alloy increased the amount of twin nucleation and decreased the size and volume fraction of the twins. Robson et al. [26] used the barrier effect of the second phase to the glide of twinning dislocation to explain the barrier effect of the second phase to twin growth. Since the aging treatment impeded the twinning deformation, the hysteresis loops became less asymmetric for T6 specimens.

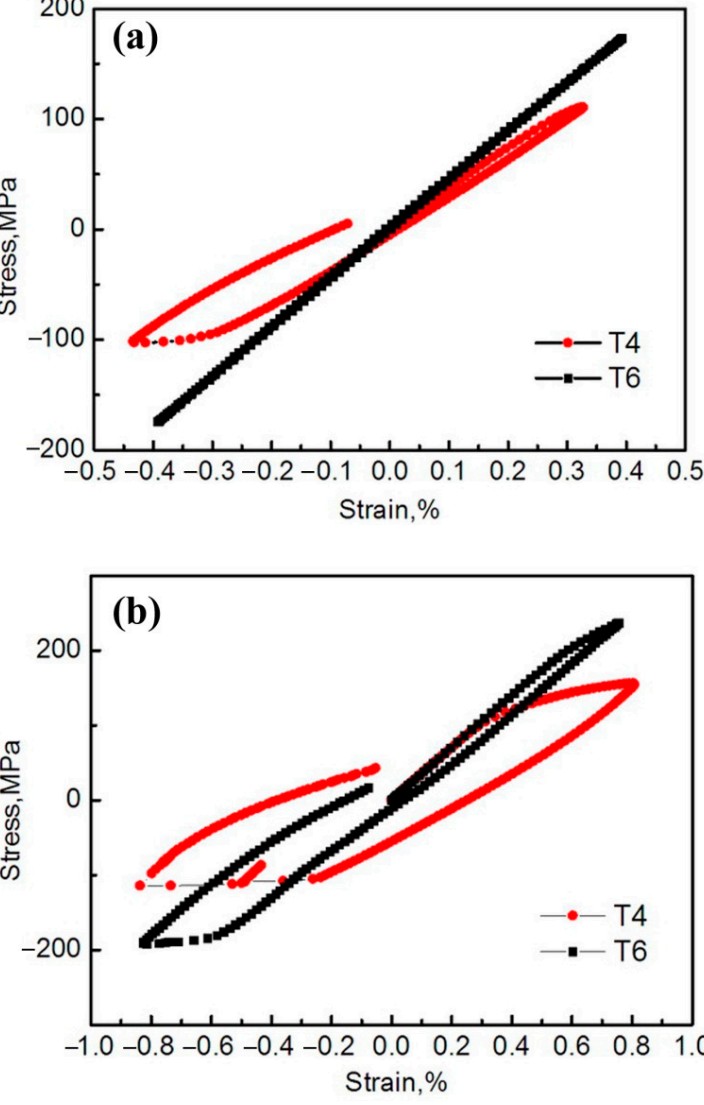

**Figure 8.** Hysteresis loops of the T4- and T6-treated specimens at strain amplitudes of (**a**) 0.4% and (**b**) 0.8% in the first cycle.

Figure 9 shows the optical microstructure close to the fracture surface of the fatigued specimen. Since no apparent plastic deformation happened, no twins were found in the T6 specimens under a strain amplitude of 0.4%.

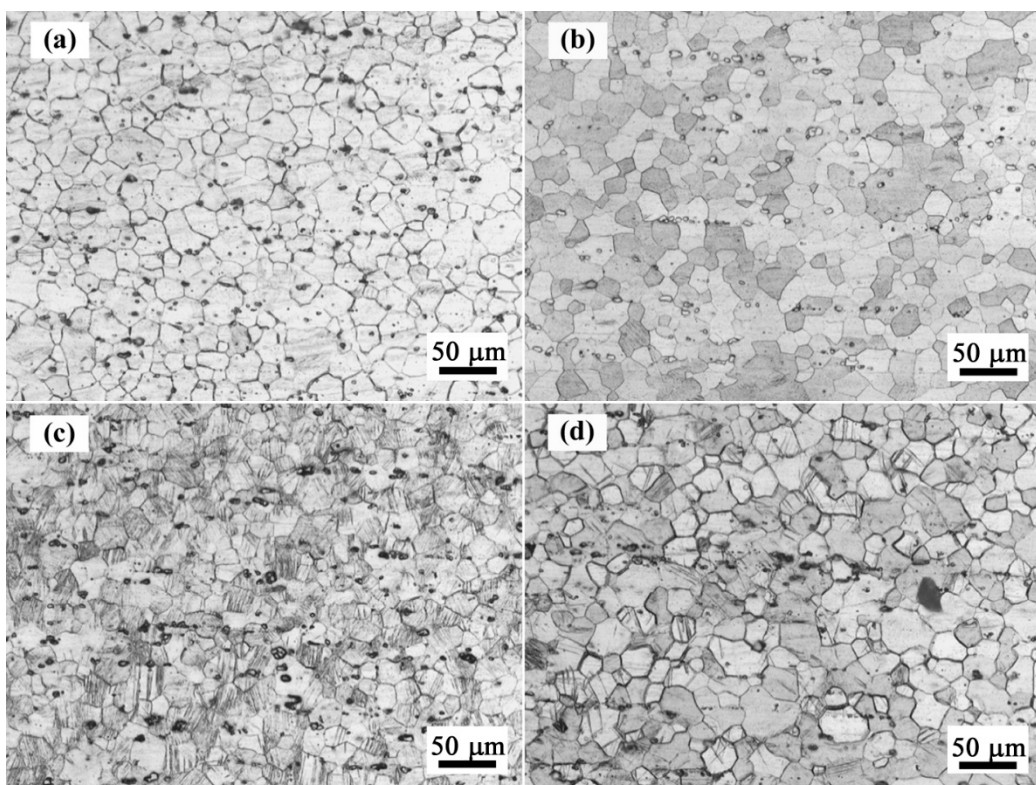

**Figure 9.** Microstructure away from the fracture surface: (**a**) T4, 0.4%; (**b**) T6, 0.4%; (**c**) T4, 0.8%; (**d**) T6, 0.8%.

In addition, residual twins could not be observed in T4 specimens, which may be the result of complete detwinning during the tensile stage, because the detwinning started just after reverse loading from the compressive loading and most of the twins generated under compression underwent detwinning due to the change in the direction of the load [27,28]. Under a strain amplitude of 0.8%, residual twins were found in both specimens with a higher volume fraction of residual twins found in the T4 specimens; thus, a longer hardening plateau was seen from the hysteresis loops in the first cycle (shown in Figure 8b).

Since the stress–strain curve determines the shape of the hysteresis loop, which affects the hysteresis energy that many fatigue models use as a fatigue parameter, it was necessary to study the evolvement of hysteresis loops with an increasing number of cycles. Non-linear stress–strain behavior was detected for both the T4 and T6 specimens during unloading at a strain amplitude of 0.8%. Figure 10 shows the instantaneous tangent modulus during tensile unloading and compressive unloading for T4 and T6 specimens during the second cycle and the half-life cycle that is smaller than the Young's modulus and decreases with an increasing number of cycles.

The decrease in instantaneous tangent modulus for T6 specimens during tensile unloading may be the result of the increasing volume fraction of residual twins. The study by Hama et al. [29] found that non-linearity became more severe for weaker textures. The formation of residual twins changed the orientation of grains and weakened the texture formed during the extrusion, which contributed to the stronger non-linearity. Figure 11 shows the instantaneous tangent modulus during the second cycle and half-life cycle tensile unloading at a strain ratio of −3 at the same strain amplitude. A higher degree of decrease in instantaneous tangent modulus was seen from the second cycle to the half-life cycle. Figure 12 shows the microstructure of T4- and T6-treated specimens near to the

fracture surface at a strain amplitude of 0.8% under different strain ratios. More twins were observed at a strain ratio of −3. Although cyclic hardening happened, the hardening rate of the T6 specimen was small and an increasing fraction of residual twins made the greatest contribution to the evolvement of the instantaneous tangent modulus.

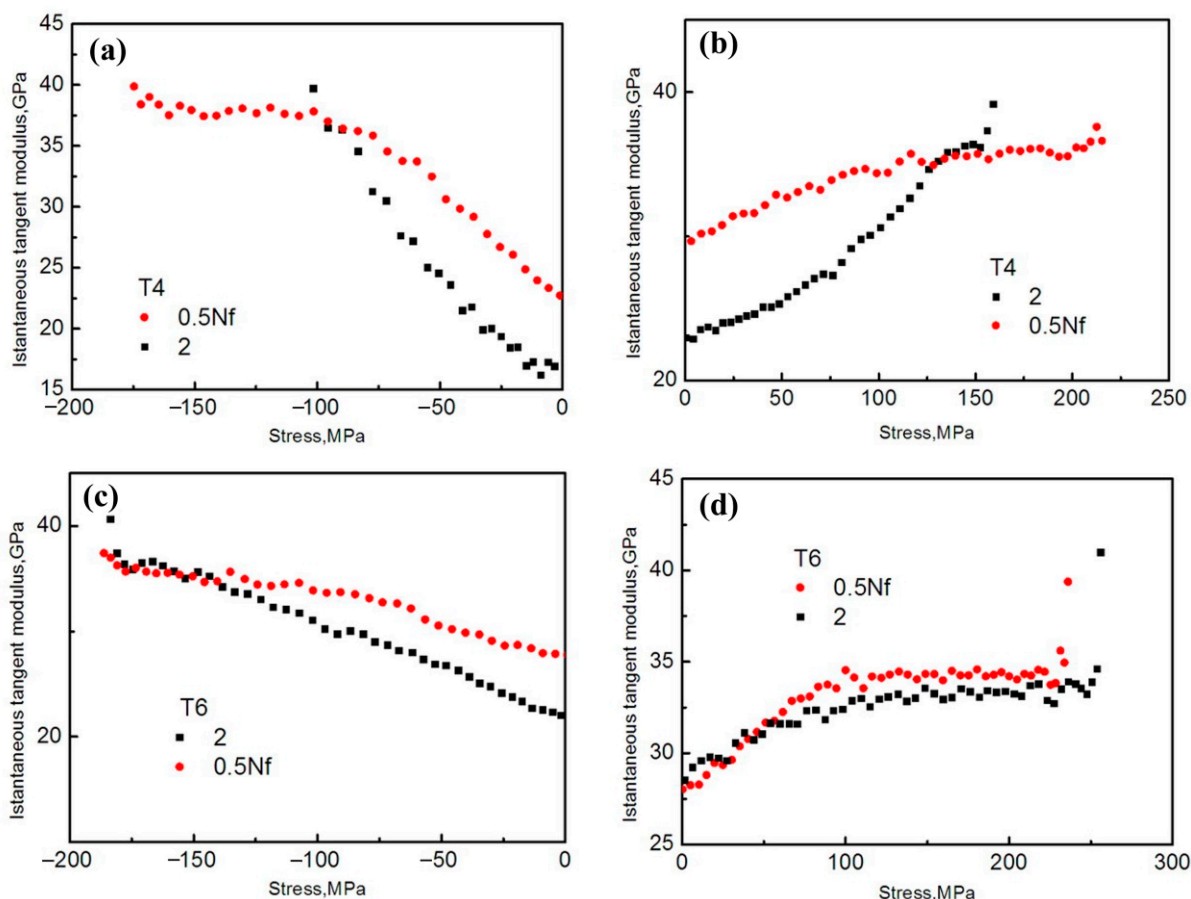

**Figure 10.** Instantaneous tangent modulus during tensile unloading (**a**,**c**) and compressive unloading (**b**,**d**) with a strain amplitude of 0.8% and a strain ratio of −1.

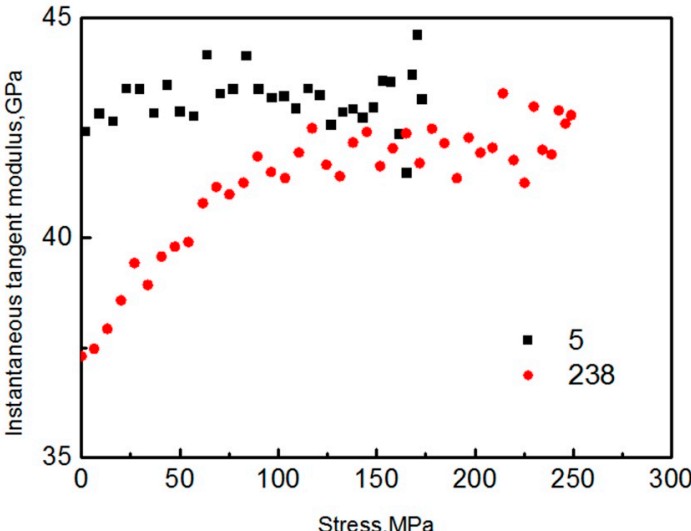

**Figure 11.** Instantaneous tangent modulus of T6 specimens during tensile unloading with a strain amplitude of 0.8% and a strain ratio of −3.

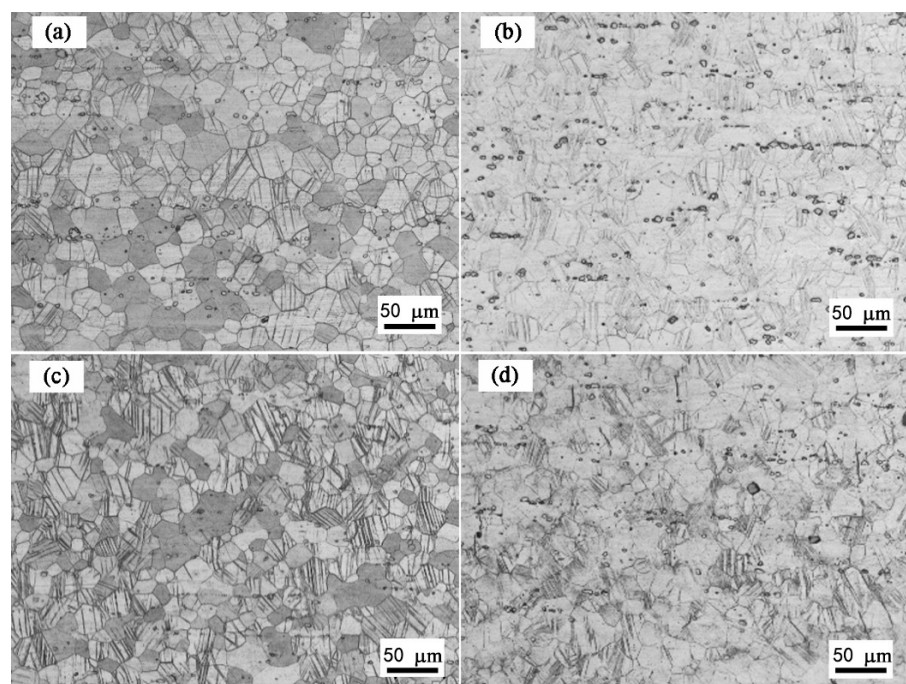

**Figure 12.** Microstructure near to the fracture surface: (**a**) T6 0.8% S = −1; (**b**) T4 0.8% S = −1; (**c**) T6 0.8% S = −3; (**d**) T4 0.8% S = −3.

Non-zero mean stresses were generated due to the asymmetric hysteresis loops. The T4 and T6 specimens showed positive and negative mean stress under a strain amplitude of 0.4%, and the absolute value of mean stress decreased with an increase in the number of cycles (as shown in Figure 13). For the T4-treated specimens, twinning deformation happened during compression under lower stress and non-basal slip happened at higher stress, which resulted in positive mean stresses. For the T6-treated specimens, symmetric hysteresis loops were developed, and the cross sectional area under compression became bigger compared with under tension, which resulted in negative mean stress. Under a strain amplitude of 0.8%, positive mean stresses were observed in both specimens. The mean stress for the T6 specimens increased as the number of cycles increased, which was the result of the greater extent of increase in the tensile peak stress. The T6 specimens exhibited a higher mean stress, although the extent of twinning deformation decreased in the second phase.

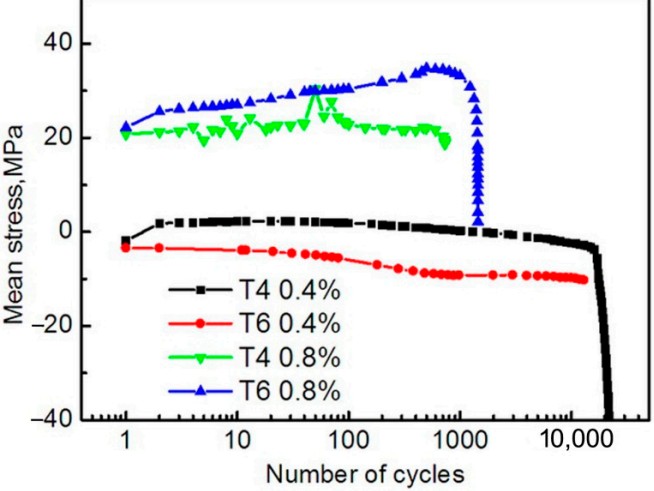

**Figure 13.** Evolvement of mean stress with increasing numbers of cycles.

Zhu et al. [30] investigated the cyclic deformation behavior of AZ80 magnesium alloy under T4 and T6 heat treatment and found that the compressive peak stresses increased and the tensile peak stresses remained constant after aging treatment and resulted in decreased tensile mean stresses, which is in agreement with the result obtained under a strain amplitude of 0.4% in the present study. However, under a strain amplitude of 0.8%, aging treatment not only increased the compressive peak stress but also increased the tensile peak stress to a higher level, which resulted in a higher positive mean stress after T6 heat treatment. Non-basal slip and twinning were the main deformation mechanisms during tension and compression, respectively, due to the texture. This result indicates that the precipitates formed during the aging treatment were more effective in hardening the non-basal slip. A similar result was found in extruded ZK60 magnesium alloy, where the mean stresses decreased after aging treatment at a strain amplitude of 0.5% and increased at higher strain amplitudes [21].

### 3.4. Fatigue Life

T4-treated specimens had a longer fatigue life under a strain amplitude of 0.4% and a shorter fatigue life under a strain amplitude of 0.8%. Since different mean stresses have been developed and positive and negative mean stress are believed to have both a detrimental and beneficial effect on fatigue life [31], it was necessary to consider the effect of different heat treatments with respect to the mean stress. The total strain energy density model was employed [32].

$$\Delta W^p + \Delta W^e = k_t \, (2N_f)^{\alpha_t} + \Delta W_0{}^t \tag{1}$$

where $\Delta W^p$ and $\Delta W^e$ are plastic strain energy density and positive elastic strain energy density, respectively; $k_t$ and $\alpha_t$ are material constants; and $\Delta W_0{}^t$ is the energy density corresponding to the fatigue strength and can be neglected when a limited fatigue life is considered. Park and Wang et al. [33,34] found that the fatigue life of AZ31 magnesium alloy can be described by this model.

For ZA81M alloy, the total strain energy density was 0.26 MJ/m$^3$ and 0.36 MJ/m$^3$ for the T4- and T6-treated specimens, respectively, under a strain amplitude of 0.4%, and 1.85 MJ/m$^3$ and 1.43 MJ/m$^3$ for the T4- and T6-treated specimens, respectively, under a strain amplitude of 0.8%. These results are in agreement with the fatigue life.

### 3.5. Fracture

Figure 14 shows that cracks in the fracture surfaces were initiated on the surface of the specimens. Under a strain amplitude of 0.4%, there was an area delineating the crack initiation area and crack propagation area, and the propagation area was relatively rough. There was one crack initiation site for the T4 specimen and two for the T6 specimen, which might be the result of the higher stress amplitude of the T6-treated specimens. Under a strain amplitude of 0.8%, there was no boundary between the crack initiation area and the propagation area.

Figure 15 shows the SEM images of the T4- and T6-treated specimen surfaces under a strain amplitude of 0.8%. Cracks shown in the T4 specimens were initiated in the second phase or along the grain boundary and propagated perpendicular to the loading direction (Figure 15a). The longer crack was initiated along the grain boundary perpendicular to the loading direction and propagated along the grain boundary and then along the slip band until the second phase.

Similar crack initiation sites were found for T6-treated specimens and the large second phase could be seen to hinder the crack propagation (Figure 15b,c). According to the TEM image of ZA81M under T6 heat treatment (Figure 4), some nano-scaled phases formed in the T6-treated ZA81M alloy, and the nano-scaled phases effectively pinned dislocations under tensile loading. However, the effects of nano-scaled phases on the fatigue properties of ZA81M alloy needs to be further researched.

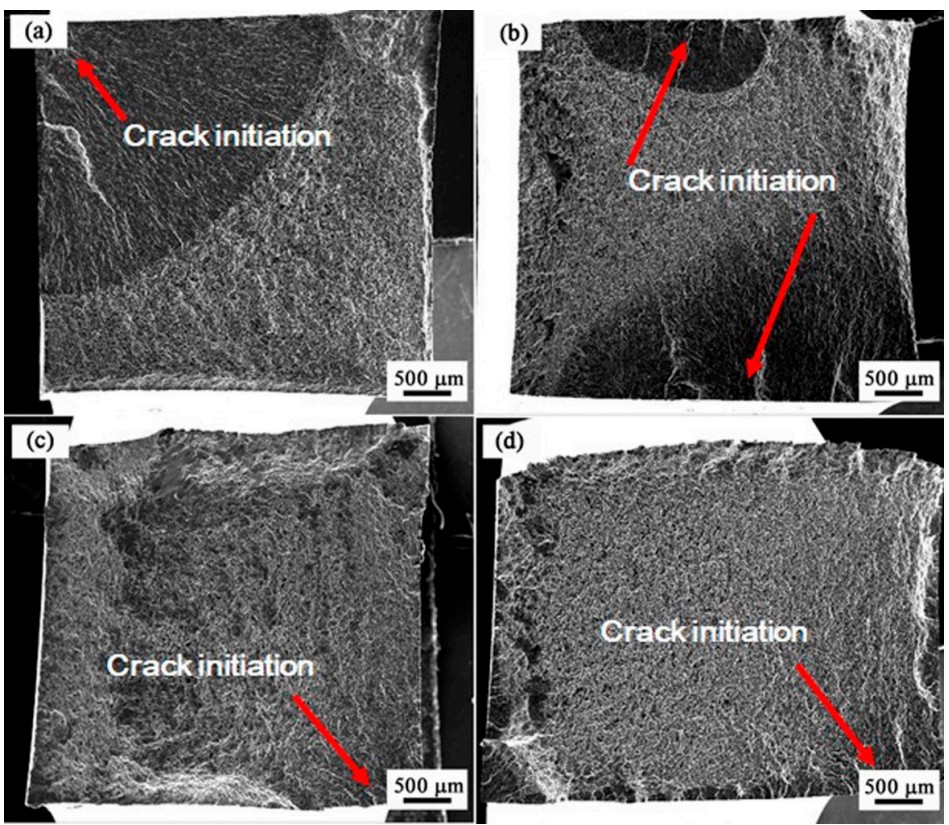

**Figure 14.** Overview of fracture surface: (**a**) T4 0.4%, (**b**) T6 0.4%, (**c**) T4 0.8%, (**d**) T6 0.8%.

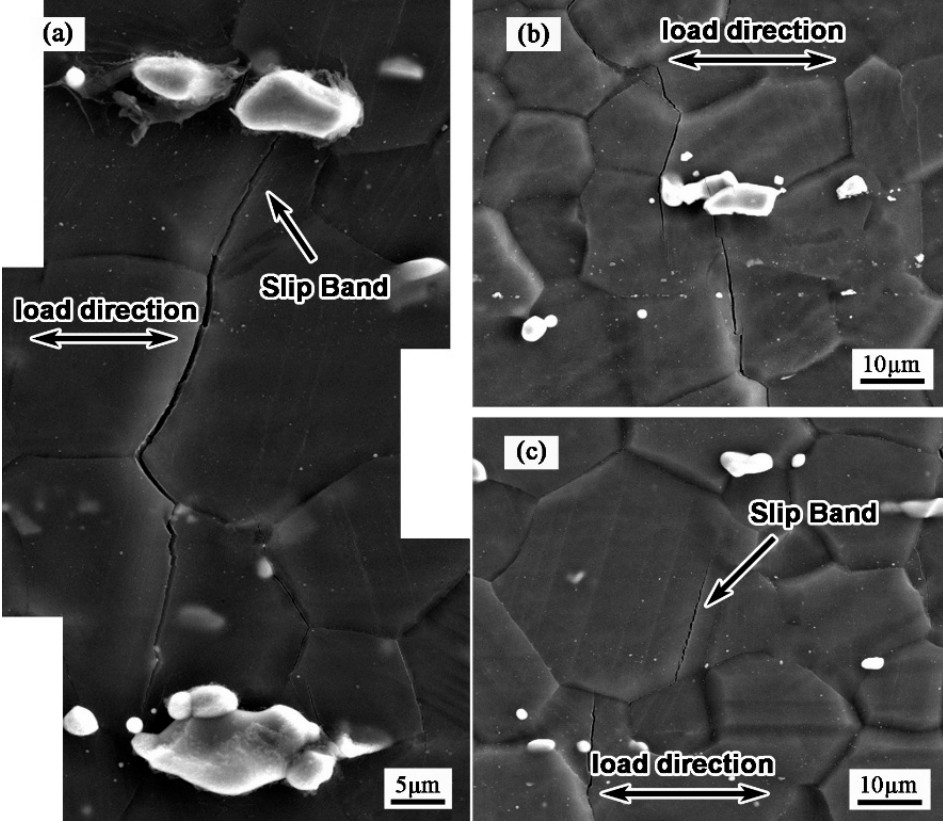

**Figure 15.** SEM images of (**a**) T4- and (**b**,**c**) T6-treated specimen surfaces under a strain amplitude of 0.8%.

## 4. Conclusions

The cyclic deformation behavior of ZA81M magnesium alloy under T4 and T6 heat treatments was investigated. The following conclusions can be drawn:

(1) In fine second-phase precipitates after aging treatment, the tensile yield strength of T6-treated specimens increased, the elongation decreased, and the ultimate tensile strength remained constant compared with T4-treated specimens.

(2) The T4- and T6-treated specimens showed cyclic hardening under strain amplitudes of 0.4% and 0.8%, respectively, with the T4 specimens showing a higher hardening rate but the stress amplitude always being higher for the T6 -treated specimens.

(3) The T6-treated specimens had a higher total strain energy density and a shorter fatigue life at a strain amplitude of 0.4% and a lower total strain energy density and longer fatigue life at a strain amplitude of 0.8% compared to the T4-treated specimens.

(4) All fatigue cracks in the T4 and T6 ZA81M alloys were initiated in the second phase or along the grain boundary perpendicular to the loading direction and propagated initially along the grain boundary and then along the slip band until they encountered the second-phase.

**Author Contributions:** Writing—original draft preparation, T.L.; Methodology, J.F., and C.L.; Investigation, C.W.; Writing—review and editing, Y.L., X.F., C.Z., and Q.H.; Funding acquisition, W.L.; Conceptualization, Y.Y. All authors have read and agreed to the published version of the manuscript.

**Funding:** This work was financially supported by the key research and development plan of Shandong province (No. 2019JZZY020329), the National Key Research and Development Program of China (No. 2021YFB3701102 and No. 2016YFB0301105), and Natural Science Foundation of Liaoning Province (No. 2020-MS-013) and DongGuan Innovative Research Team Program (No. 2020607134012).

**Institutional Review Board Statement:** Not applicable.

**Informed Consent Statement:** Not applicable.

**Data Availability Statement:** Not applicable.

**Conflicts of Interest:** The authors declare that they have no known competing financial interests or personal relationships that could appear to influence the work reported in this paper.

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
