# Peer review of "Effect of Heat Treatment on the Cyclic Deforming Behavior of As-Extruded ZA81M Magnesium Alloy"

_metals, doi:10.3390/met12010146_

Round 1

Reviewer 1 Report

The work addresses the effect of heat treatment on the cyclic deformation behavior of as-extruded ZA81M magnesium alloy. The two heat treatment conditions applied to the as-extruded ZA81M alloy were appplied: solution treatment (T4 state) and solution treatment plus artificial aging. Interesting results were presented including fatigue tests.

Remarks:

1/ Introduction is too short. The deeper analysis myst be performer focusing on recent fatigue tests of magnesium alloys. The real state of the art must be provided on fatigue testing methods, major problems, etc.

2/ Is it a laboratory sample or an industrial ingot. More details are needed concerning dimention of the slab, etc.

3/ How did you selcet heat treatment parameters ? Previous studies ?

4/ The methodology is very poor described. The details concernin the fatigue tests and microscopis research must be provided for the aim to be able to reproduce the experiment.

5/ The novelty of the study must be crearly emphasized.

6/ The dewinning proces is interesting and has be negleted here. The explanation of the proces is required here.

7/ The scientific discussion is limited here. The improvement of the discussion with the literature data is needed to consider the manuscript as a scientific work.

8/ The numer of references is low. There are a lot of published papers in this topic, which should be references and properly disscussed.  

Reviewer 2 Report

  1. The introduction is very poor and has to be developed.The article concerns cyclic deformation of ZA81M magnesium alloy.In this respect, many works for different magnesium alloys are published.The authors should refer to the results of other works and based on it describe precisely the aim of this paper.
  2. T4 and T6 state alloys are mentioned in the introduction, but there is no information about them. What do T4 and T6 states mean?
  3. Clearly state the novelty aspects of the paper;
  4. The following corrections are needed:
  • “Figure 1. Shapes of specimens (mm): (a) tensile test specimens; (b) fatigue specimens.” should be Figure 1. Shapes of specimens (mm): (a) tensile test specimen; (b) fatigue specimen.
  • „equiaxed grains about 15mm” should be: equiaxed grains about 15microns. Moreover what does it mean? Average grain size diameter? If yes, how was it calculated?
  • „Fig-ure.3” should be Figure 3,
  1. “the alloy in T4 and T6 states are composed of -Mg, MgZn2 and MgZnCu phase”. These phases should be identified not only by XRD but also should be presented in figs showing structural results.
  2. What type of phase is presented in Fig. 4? This phase should be identified using electron diffraction patterns and described in the paper.
  3. Clarification of the statement “…away from the fracture surface” is needed.
  4. Cracked initiation sites should be marked in Figure 14
  5. Conclusions should highlight the new findings of the paper in comparison to the state of the art.

Reviewer 3 Report

Dear Authors,

The paper is very interesting. I do not know why you have decided to work with fatigue tests across the zero. Generally this will generate more complex tests and the risk of shifting during the zero-transition, so it should be interetsing thatyou state that no shifting has been detected during the whole tests, nevetheless the results are well explained.

Some Comments:

Line 38: Please re-edit this sentence. It is obvious that the heat treatment had a significant effect…..”.  Try to state that:  “… an optimised heat treatment permits to obtain improved performances …ie…:….”

Line 97 -104 and Figure 4: Try to link these sentences for the figure 4. Add on the figure the label for the precipitates and also the alpha matrix axis direction.

Line 163: You state that the hardening is the result for the twinning deformation: I can agree on the twinning…but what are the reason or the literature references that can be the base for this assumption? Please add and discuss in a more complete way, explain the mechanism for a stable plateau Vs twinning?

Figure 8: quote (a) and  (b)…in the figure caption.

The initial hysteresis loop is interesting, but please add also the stable loop or state that was never obtained a stable loop.

Fig.15: please ad details about this figure, it is not clear if it is SEM images or what ?

Conclusions: “All the cracks start at grain boundary or second phases”, please add details about what orientations have the cracks with respect to loading axis, the cracks are isolated? There are preference sites for crack nucleation depending  from the second phases dimensions? Do you have carry on some TEM close to the fractures?

Round 2

Reviewer 1 Report

The authors improved the manuscript according to my comments. The methodology is enriched and the introduction and discussion are also corrected / modified. The comments are implemented to the current version of the manuscript.

Reviewer 2 Report

Authors improved the manuscript according to suggestions.